# Single-Molecule Insights into ATP-Dependent Conformational Dynamics of Nucleoprotein Filaments of *Deinococcus radiodurans* RecA

**DOI:** 10.3390/ijms21197389

**Published:** 2020-10-07

**Authors:** Aleksandr Alekseev, Galina Cherevatenko, Maksim Serdakov, Georgii Pobegalov, Alexander Yakimov, Irina Bakhlanova, Dmitry Baitin, Mikhail Khodorkovskii

**Affiliations:** 1Peter the Great St Petersburg Polytechnic University, St Petersburg 195251, Russia; alex35093@gmail.com (A.A.); gcherevatenko@gmail.com (G.C.); maximilianka2392@gmail.com (M.S.); yaleks@nanobio.spbstu.ru (A.Y.); khodorkovskii@mail.ru (M.K.); 2Department of Molecular and Radiation Biophysics, Petersburg Nuclear Physics Institute (B.P. Konstantinov of National Research Centre ‘Kurchatov Institute’), Gatchina 188300, Russia; ibakh2003@mail.ru (I.B.); dimabaitin@yahoo.com (D.B.); 3Institute of Cytology, Russian Academy of Sciences, St. Petersburg 194064, Russia

**Keywords:** homologous recombination, DNA repair, RecA, *Deinococcus radiodurans*

## Abstract

*Deinococcus radiodurans* (Dr) has one of the most robust DNA repair systems, which is capable of withstanding extreme doses of ionizing radiation and other sources of DNA damage. DrRecA, a central enzyme of recombinational DNA repair, is essential for extreme radioresistance. In the presence of ATP, DrRecA forms nucleoprotein filaments on DNA, similar to other bacterial RecA and eukaryotic DNA strand exchange proteins. However, DrRecA catalyzes DNA strand exchange in a unique reverse pathway. Here, we study the dynamics of DrRecA filaments formed on individual molecules of duplex and single-stranded DNA, and we follow conformational transitions triggered by ATP hydrolysis. Our results reveal that ATP hydrolysis promotes rapid DrRecA dissociation from duplex DNA, whereas on single-stranded DNA, DrRecA filaments interconvert between stretched and compressed conformations, which is a behavior shared by *E. coli* RecA and human Rad51. This indicates a high conservation of conformational switching in nucleoprotein filaments and suggests that additional factors might contribute to an inverse pathway of DrRecA strand exchange.

## 1. Introduction

The radioresistant bacterium *Deinococcus radiodurans* and other members of the Deinococcaceae show an outstanding capacity to cope with high dosage of ionizing radiation and other DNA-damaging agents, such as desiccation, ultraviolet radiation, and diverse genotoxic chemicals [1,2,3,4,5]. The extreme radiation resistance of *D. radiodurans* has been attributed to a strong protection of the proteins from oxidative damage [6,7] and a very robust DNA repair system [3,8].

The RecA protein is key for the extreme resistance of *D. radiodurans* [8,9]. DrRecA belongs to a highly conserved family of bacterial homologous recombination proteins that promote the error-free repair of DNA damage [10,11]. *D. radiodurans* expressing RecA variants defective in recombination are highly sensitive to ionizing radiation [12,13]. Apart from homologous recombination, DrRecA was reported to be crucial for the extended synthesis-dependent strand annealing (ESDSA), which is a unique mechanism of fragmented chromosome segments assembly at the early phase of DNA repair in *D. radiodurans* [8,14]. *D. radiodurans* is immutable by ultraviolet radiation due to the error-free reparation of ultraviolet-induced DNA damage [15], whereas an error-prone pathway for the repair of such DNA damage was found in *D. deserti* [16], which is another species belonging to Deinococcaceae.

DrRecA shares only 56% of amino acid sequence identity with *E. coli* RecA (EcRecA) [17]. The expression of *Shigella flexneri* RecA, identical to that of *E. coli*, in *recA*-defective *D. radiodurans* leads to the loss of radioresistance, whereas the expression of DrRecA was reported to be toxic to *E. coli*, indicating the unique functional properties of DrRecA [12,18]. Prokaryotic RecA family proteins as well as eukaryotic Rad51, Dmc1, and archaeal RadA act through the assembly of long right-handed filaments on single-stranded trails (ssDNA) of damaged DNA in an ATP-dependent manner [19,20,21,22,23]. Successively, nucleoprotein filaments search for homologous double-stranded DNA (dsDNA) and perform strand exchange in case of a sufficient degree of homology [10,24]. Strikingly, DrRecA was reported to act via a reverse pathway, initiating strand exchange from dsDNA [25]; however, the molecular mechanism underlying this unique property of DrRecA is not fully understood.

Direct comparison of DNA binding by *D. radiodurans* and *E. coli* RecA using defined short dsDNA and ssDNA oligomers revealed that in the absence of nucleotide cofactors, DrRecA binds dsDNA over an order of magnitude more tightly compared to EcRecA [26]. In addition, DrRecA showed altered binding hierarchy, with higher affinity to DNA when no nucleotide cofactors are present than in the presence of ATPγS, which is a nonhydrolyzable ATP analogue. In contrast, EcRecA binds DNA more tightly in the presence of ATPγS. Both proteins showed the lowest affinity to DNA in the presence of ADP. Efficient DrRecA binding to short dsDNA fragments might play an important role in the ESDSA mechanism. However, the functioning of RecA in homologous recombination requires the formation of long nucleoprotein filaments in the presence of ATP [27].

Extensive structural studies of the *E. coli* RecA filaments revealed the existence of two conformational states. The active state can be formed on ssDNA and dsDNA in the presence of ATP or its nonhydrolyzable analogs, which is characterized by a stoichiometry of 1 monomer per 3 nt of DNA and a helical pitch in the range of 89 to 95 Å [22,28,29]. Alternatively, ADP or no nucleotide produces a more compressed inactive state with helical pitch values ranging from 70 to 76 Å [30,31]. To date, only a structure of DrRecA filament in inactive conformation has been resolved without DNA and in the presence of ATPγS with a helical pitch of 67 Å, making it the most compressed of any RecA filament structure [17]. The overall structures of *D. radiodurans* and *E. coli* RecA were reported to be similar; however, one of the key differences included an increased positive electrostatic surface potential along the central groove of the filament, which might dictate the preference to dsDNA over ssDNA.

DrRecA has been shown to be a subject for the phosphorylation by RqkA, which is a cognate DNA damage-responsive quinoprotein kinase and a feature uncommon to other known RecA proteins [32]. Phosphorylation affected the preference of DrRecA to dATP over ATP in strand exchange reactions in vitro and radioresistance in vivo. Recent molecular dynamics modeling suggested that DrRecA phosphorylation may affect its dynamics and conformational plasticity and modulate its nucleotide binding and DNA binding preferences [33]. Single-molecule observations of the *E. coli* RecA and human Rad51 nucleoprotein filaments dynamics revealed that filaments formed on ssDNA may reversibly interconvert between active and inactive states in response to ATP-binding and hydrolysis [34,35,36,37]. ATP hydrolysis is essential for the processes of homology search and strand exchange, which suggests the importance of dynamic conformational switching between the two states [38,39,40].

Single-molecule insights into dsDNA binding by DrRecA provided an important observation of the faster nucleation and slower elongation of DrRecA compared to EcRecA [41]. Furthermore, direct comparison of the mechanical properties of the nucleoprotein filaments formed on dsDNA revealed that DrRecA filaments are shorter and more flexible, which is a feature that might support the efficient repair of numerous concurrent DNA double-strand breaks [42]. However, little is known about the conformational dynamics of the DrRecA nucleoprotein filaments and its response to ATP hydrolysis. In this work we utilize single-molecule DNA manipulation using optical tweezers to accurately address the behavior of DrRecA filaments formed on both ss- and dsDNA and address the role of ATP binding and hydrolysis in the conformational switching of DrRecA.

## 2. Results

### 2.1. Assembly of DrRecA Filament on Single-Stranded DNA

In order to examine the interaction of DrRecA with ssDNA, we initially followed the change in length of individual DNA tethers in the presence of DrRecA and ATP using an approach combining DNA manipulation by optical tweezers and a multichannel microfluidic flow system (uFlux, Lumicks) published elsewhere [42,43,44,45]. The formation of a nucleoprotein complex by RecA-family proteins leads to a drastic change in the mechanical properties of the DNA molecule at the core of the complex [22]. To register the assembly of the DrRecA–ssDNA filament, a single ssDNA molecule was introduced to the channel containing 1 µM DrRecA and 1 mM ATP, and the change in the end-to-end distance was monitored under a constant applied tension of 12 pN that facilitated the removal of ssDNA secondary structures. Experiments were performed at 22 °C. The binding of DrRecA resulted in an increase in the end-to-end distance of the tether (Figure 1A), reaching a maximum value of 5.37 ± 0.06 µm (*n* = 12). That counts for only 42.6 ± 1.6% elongation in respect to the contour length of corresponding dsDNA, which is shorter than the 48% extension reported for EcRecA filaments assembled on ssDNA [34].

### 2.2. Interconvertibility of Active and Compressed States of DrRecA–ssDNA Filament

We further addressed the dynamics of assembled DrRecA–ssDNA filaments by transferring them between the ATP-containing channel and the ATP-free one. During transition, a constant force of 2-3 pN was applied to the filament, and its length was monitored (Figure 1B). Upon transfer into the ATP-free buffer, the filament length sharply decreased by about 30% at first and was relatively constant during further incubation. The subsequent introduction of compressed DrRecA–ssDNA filaments back to the ATP-containing channel led to the elongation by 25.0 ± 2.8% (N = 25). Importantly, this observation was registered in the absence of free DrRecA, excluding its additional binding; hence, the tether elongation is due to a conformational transition of the DNA-bound DrRecA from a compressed form to the stretched one that is triggered by ATP binding. Interestingly, switching between stretched and compressed forms could be observed multiple times on the same tether in the absence of free DrRecA, indicating the direct interconvertibility of the two conformations (Figure 1B). However, two points should be noted. First, the dynamics of the transition from compressed into the stretched state is rather complicated: an initial sharp increase in the filament length is followed by slow additional growth in the ATP channel that may reflect some reorganization of the filament structure after the major part is switched to the active state. Second, reintroduction of the compressed filament to the ATP channel does not lead to the full restoration of its initial length, which is possibly due to a dissociation of a small part of the filament in the course of conformational rearrangements.

Existence of the two interconvertible states of the DrRecA–ssDNA filaments depending on the presence of ATP suggests that ATP hydrolysis promotes a conformational change from the stretched to the compressed form that can be reversed by subsequent ATP binding. However, the filaments exhibited the stretched conformation when incubated in the ATP-containing channel while ATP hydrolysis was constantly occurring throughout the filament. To verify the existence of local patches of the compressed form within the long DrRecA–ssDNA filaments under continuous ATP hydrolysis, we transferred preassembled filaments from the ATP-containing channel to the one where ATP was replaced by its nonhydrolyzable analog, ATPγS, thus fixing all DrRecA monomers in the ATP-bound form. The substitution of ATP with ATPγS resulted in a small but clear increase in the tether length (Appendix A) while no free DrRecA was present, indicating that in the presence of ATP, a small fraction of transiently formed compressed state is present within the long filament. This observation suggested that under the constant supply of ATP, the DrRecA–ssDNA filaments retain an overall stretched conformation while locally occurring events of ATP hydrolysis might promote local conformational transitions between stretched and compressed states, resulting in a dynamic heterogeneous structure.

### 2.3. Force–Extension Behavior of Two States of DrRecA–ssDNA Filament

The mechanical properties of DrRecA–ssDNA filaments were further addressed by stretching and relaxing the tether while simultaneously registering its length and applied tension. A force–extension curve of DrRecA–ssDNA in the presence of ATP indicates a stiff structure of the complex formed as result of DrRecA polymerization on ssDNA. The initial part of the curve is characterized by about zero measured force until the end-to-end distance reaches the contour length, followed by a steep increase in the force upon further pulling (Figure 2). Such force–extension behavior of DrRecA–ssDNA filaments is similar to that previously reported for EcRecA filaments on both ssDNA and dsDNA [34,46] and DrRecA filaments on dsDNA [42].

The compressed state of the DrRecA–ssDNA filament exhibits intrinsic force–extension behavior (Figure 2). Upon stretching, the initial rise in force occurs at a lower end-to-end distance compared to that of the extended ATP-bound state. However, at forces higher than 7.9 ± 0.7 pN (N = 9), a sharp change in the slope of the force–extension curve is observed, which is characterized by the significantly enhanced extensibility of the tether. An abrupt transition at similar stretching forces has been previously reported for the inactive form of EcRecA–ssDNA filaments [34,46]. However, unlike the constant force plateau around 8 pN measured for EcRecA–ssDNA filaments in the presence of ADP [46], the stretching of inactive DrRecA–ssDNA filaments at forces higher than 8 pN resulted in a continuous force increase, approaching a force–extension behavior of bare ssDNA (Appendix A) similar to [34]. Although the nature of such a force-induced transition cannot be resolved solely based on force–extension analysis, extension and relaxation of the inactive DrRecA–ssDNA filament follow almost identical curves, likely reflecting that similar a DNA–protein complex exists during the stretching–relaxation cycle.

### 2.4. Dynamics of DrRecA Filament on Double-Stranded DNA

In contrast to ssDNA, the assembly of DrRecA filaments on dsDNA under identical conditions was constrained. Applying tension of 12 pN did not initiate the assembly of the filament. Therefore, the force was increased in small steps until the DrRecA polymerization on dsDNA was detected (Figure 3A). Efficient DrRecA nucleation and filaments assembly on dsDNA were observed under an applied tension of 50 pN at 22 °C. Under tension of 3 pN, assembled DrRecA–dsDNA filaments were stable when both free DrRecA and ATP were presented in solution, but they exhibited a shorter length compared to DrRecA–ssDNA filaments (Figure 3B). In the channel containing ATP and no free DrRecA a significant decline in filament length was observed. Moving DrRecA–dsDNA filaments to the ATP-lacking solution led to the fast shrinking of the tether length down to 3.38 ± 0.05 µm (*n* = 10), which corresponds to the length of bare dsDNA. To test whether the decrease in the tether length is due to DrRecA dissociation or a conformational change as in the case of the filaments formed on ssDNA, the tether was moved back to the channel containing ATP. This time, no change in the tether length was observed, indicating that in the absence of free ATP, DrRecA completely dissociated from dsDNA.

It is noteworthy that the elimination of ATP led to fast DrRecA dissociation from dsDNA, even in the presence of free DrRecA in a solution (Appendix A).

Considering that room temperature possibly is not a favorable condition for the proper assembly of DrRecA–dsDNA filaments [42,47], we further tested dsDNA–DrRecA interaction at a higher temperature of 37 °C and under lower Mg^2+^ ions concentration (1.5 mM MgCl_2_). Under these conditions, the DrRecA polymerization was less constrained and proceeded under a lower applied tension of 20 pN (Figure 3C). However, DrRecA–dsDNA filaments showed the same fast and complete disassembly upon ATP removal (Figure 3D), resulting in no change in the length of the tether upon reintroduction into the ATP-containing channel. Under both tested conditions, DrRecA–dsDNA filaments exhibited the length of 4.38 ± 0.21 µm (*n* = 22) at 3 pN in the presence of both DrRecA and ATP, indicating only a partial coverage of DNA.

### 2.5. Compression of DrRecA Filaments Is Induced by ATP Hydrolysis

To further address the role of ATP hydrolysis in the observed behavior of the filaments on both ss- and dsDNA, we assembled DrRecA filaments in the presence of 1 mM ATPγS. Then, we followed the change in the filaments length during incubation and upon transitions between the channels containing respectively 1 µM DrRecA and 1 mM ATPγS, 1 mM ATPγS without DrRecA, and buffer without both DrRecA and ATPγS (Figure 4). The lengths of DrRecA–ssDNA and DrRecA–dsDNA filaments after assembly were close to each other: 5.34 ± 0.10 µm (*n* = 3) and 5.32 ± 0.04 µm (*n* = 4) respectively under applied tension of 3 pN. DrRecA filaments on both ss- and dsDNA exhibited great stability in the channel containing 1 mM ATPγS and no DrRecA. Transition to the channel lacking ATPγS did not lead to any compression of the filaments. However, subsequent incubation in the absence of ATPγS revealed a slow decline in the length of the DrRecA–ssDNA filament, whereas the length of the DrRecA filament formed on dsDNA was almost unchanged. This fact could be attributed to a very slow proceeding of DrRecA dissociation from ssDNA. Previously, a slow dissociation of EcRecA was reported from filaments formed in the presence of ATPγS on ssDNA but not dsDNA [37].

## 3. Discussion

In this work, we examined the dynamics of interaction of DrRecA with both ssDNA and dsDNA using a single-molecule approach. We report that DrRecA filaments assembled on ssDNA are stable in the absence of free DrRecA and undergo a fast compaction upon transition into the ATP-lacking environment. Subsequent reintroduction into the ATP-rich environment restores the overall stretched form. Relative elongation of the DrRecA-ssDNA filaments upon transition from the compressed to the stretched state is 25.0 ± 2.8%, which is in a good agreement with the ratio between reported helical pitch values of ≈ 95 and ≈ 76 Å of the active and inactive EcRecA filament structures [22,30]. Hence, our data provide direct evidence that ATP hydrolysis induces fast conformational transition of the DrRecA–ssDNA filament from the active state into the inactive one, which can be reversed by the subsequent binding of new ATP, while DrRecA dissociation from ssDNA proceeds very slowly. Direct transitions between two conformational states induced by ATP binding and hydrolysis were previously reported for nucleoprotein filaments formed on ssDNA by the *E. coli* RecA and human Rad51 [34,35,36,37]. The fact that DrRecA–ssDNA filaments exhibit two interconvertible states depending on the presence of ATP indicates a particularly high conservation of conformational switching throughout nucleoprotein complexes formed by the RecA family proteins on single-stranded DNA. 

In contrast, we could not detect a stable inactive conformation of DrRecA filaments on dsDNA. Under the same experimental conditions, assembly of the DrRecA–dsDNA was constrained and initiated at a much higher stretching force (50 pN at room temperature, 20 pN at 37 °C) compared to the nucleoprotein filaments assembly on ssDNA (12 pN at room temperature), indicating that the energy barrier for the DrRecA nucleation on dsDNA is higher than on ssDNA. Previously, it was established that in the presence of ATPγS, DrRecA shows greater affinity for ssDNA over dsDNA, while in the absence of the nucleotide cofactors, the affinity of DrRecA for ssDNA increases with the length of DNA and exceeds the affinity of DrRecA for dsDNA already when the oligomers are longer than 13-30 nucleotides [26]. These data represent the initial binding of DrRecA to DNA and are in agreement with our observation that the DrRecA filaments formation on long ssDNA is favored over dsDNA. The length of DrRecA–dsDNA filaments under ATP-hydrolysis conditions was shorter than that of DrRecA–ssDNA filaments and stable only when both DrRecA and ATP were present. The removal of DrRecA led to a fast decrease in the filament length, suggesting a fast dissociation rate of DrRecA from dsDNA. Transition into the ATP-free environment led to an instant irreversible shortening of the tether down to the length of bare dsDNA, corresponding to the complete dissociation of DrRecA from dsDNA.

To further test the role of ATP hydrolysis in observed conformational transitions of DrRecA nucleoprotein filaments, we studied the behavior of the filaments assembled in the presence of ATPγS. The length of the DrRecA filaments on ssDNA was comparable to the ones formed in the presence of ATP; however, no compression of the filaments was observed after the removal of free DrRecA and nucleotides, hence providing direct evidence that ATP hydrolysis is essential for dynamic conformational switching between active and inactive states of the DrRecA–ssDNA filament. Interestingly, DrRecA–dsDNA filaments assembled in the presence of ATPγS were significantly longer than under ATP-hydrolysis conditions, reaching the length comparable to DrRecA–ssDNA filaments. This indicates the similarity of the structure of DrRecA filaments formed on ss- and dsDNA under no ATP hydrolysis conditions. Observation of the longer filaments on dsDNA in the absence of ATP hydrolysis provides further evidence of a higher rate of ATPase-induced DrRecA dissociation from dsDNA, which results in incomplete coverage of the dsDNA molecule when ATP hydrolysis is permitted. The removal of free DrRecA and nucleotides had no effect on the length of DrRecA–dsDNA filaments assembled in the presence of ATPγS, revealing its even greater stability than DrRecA–ssDNA filaments and confirming the essential role of ATP hydrolysis in DrRecA dissociation from dsDNA.

Overall, our results provide new insights into the dynamics of DrRecA–DNA interactions. We established that under identical conditions, DrRecA may form stable nucleoprotein filaments on both single-stranded and double-stranded DNA; however, exchange between free DrRecA and DrRecA bound within the filament on dsDNA is much more dynamic than for the DrRecA filaments on ssDNA. Constantly proceeding cycles of ATP binding and hydrolysis promote DrRecA binding and dissociation from dsDNA, whereas on ssDNA, DrRecA remains bound and undergoes a local conformational switching between stretched (active) and compressed (inactive) filament states, resembling the previously reported behavior of EcRecA filaments on ssDNA [37]. Recently, molecular dynamics simulation provided structural evidence of the importance of active–inactive conformational transitions resulting in local heterogeneity in the structure of EcRecA filaments for the process of homology recognition and strand exchange [48]. Considering the fact that both DrRecA and EcRecA show similar conformational switching triggered by ATP hydrolysis for the filaments formed on ssDNA but not on dsDNA, the ability of DrRecA to promote reverse strand-exchange seems even more puzzling. We believe that further single-molecule studies of the role of DrRecA phosphorylation in the nucleoprotein filaments dynamics and strand-exchange reaction will help to shed more light on that unique mechanism.

## 4. Materials and Methods

### 4.1. DNA Construct and Proteins

To obtain double-stranded DNA molecules with biotinylated 5′- and 3′- ends of the same strand, a plasmid vector prl574 with the insertion of the rpoC gene (11,344 bp in total) was digested with XbaI and SacI restriction enzymes (Thermo Fisher Scientific, Waltham, MA, USA). The product of double digestion was ligated with a 50-fold excess of oligonucleotides 5′-XXXXXCAGTCCAGCT-3′ and 5′-CTAGCGAGTGXXXXX-3′, where X denotes a biotin-labeled nucleotide (Alkor Bio, St. Petersburg, Russia). Short complementary oligonucleotides 5′-GGACTG-3′ and 5′-CACTCG-3′ (Alkor Bio, St. Petersburg, Russia) were added to the reaction in order to increase ligation efficiency [49]. A reaction was carried out at 22 °C for 2 h and heat-inactivated at 65 °C for 20 min. The final DNA construct (≈ 11 Kbp long, corresponding to 3.76 µm contour length) was purified using Bio-Gel P-30 size-exclusion spin-column (Bio-Rad Laboratories, Hercules, CA, USA).

For experiments with ssDNA, the same DNA construct with biotinylated ends was used. The biotinylation of 5′- and 3′- ends of the same strand allowed generating an ssDNA substrate by a force-induced melting technique during single-molecule assay [50]. To this end, a captured duplex DNA molecule was stretched with a force above 80 pN for ten seconds, which resulted in the melting of a dsDNA molecule and dissociation of the unlabeled strand (Appendix A). After relaxation of the tether, an additional cycle of force–extension measurements was performed to verify that dsDNA was fully converted to ssDNA.

Wild-type DrRecA protein was purified as previously described [51].

### 4.2. Optical Tweezers Setup

A custom dual-trap optical tweezers setup built around an upright fluorescent microscope (AxioImager.Z1, Carl Zeiss, Oberkochen, Germany) was used as described previously [42,52]. In brief, a ND:YVO4 1064 nm laser beam (Spectra-Physics, Mountain View, CA, USA) was split in two using a polarizing beam splitter cube and focused with an oil immersion lens (LOMO 100X, NA 1.32, St. Petersburg, Russia) to generate two orthogonally polarized optical traps. The x–y position of one of the traps was operated by the mirror mounted on a piezo platform (S-330.80L, Physik Instrumente, Karlsruhe, Germany). The images of the trapped beads were obtained with an EMCCD camera (Andor Technology, iXon Ultra 897, Belfast, UK) and further processed for real-time measurements of DNA-tether length and applied tension with 30 ms time resolution. Force–clamp and force–extension measurements were performed using custom software developed in LabView.

### 4.3. Single-Molecule Assay

Single-molecule DNA manipulation was performed in the five-channel microfluidic flow cell (u-Flux, LUMICKS B.V., Amsterdam, Netherlands). The flow cell was fed with following solutions: 1st channel—2.1 µm streptavidin-coated polystyrene beads (0.01%, Spherotech, Lake Forest, IL, USA), 2nd—30 pM of biotinylated dsDNA; 3rd—buffer solution; 4th—1 mM ATP (Sigma-Aldrich, Saint Louis, MO, USA); 5th—1 µM DrRecA, 1 mM ATP. Channels containing ATP were supplemented with an ATP regeneration system: 10 U/mL pyruvate kinase, 0.1 mM phosphoenolpyruvate (Sigma-Aldrich, Saint Louis, MO, USA). Buffer solution used in all channels was 25 mM Tris-HCl (pH = 7.5) (AMRESCO, LLC, Solon, OH, USA), 5 mM MgCl_2_ (AppliChem GmbH – An ITW Company, Darmstadt, Germany). All experiments were performed at 22°C, unless otherwise stated.

The first three channels were used for capturing two beads, attaching single dsDNA molecule to them, and the generation of ssDNA by force-induced melting. The polymerization of DrRecA on an ssDNA or a dsDNA was performed in the 5th channel by operating optical tweezers in a force–clamp mode.

## Figures and Tables

**Figure 1 ijms-21-07389-f001:**
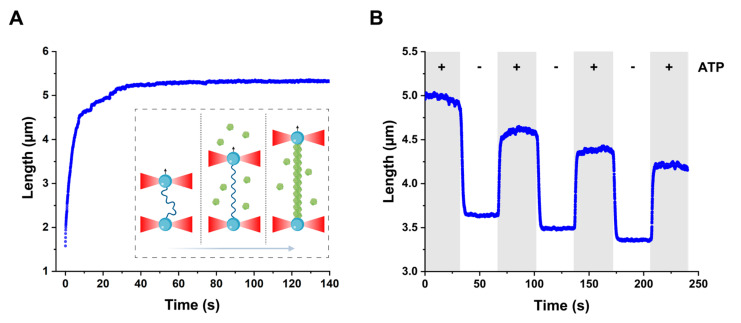
Dynamics of DrRecA–ssDNA filaments length. (**A**) DrRecA binding leads to ssDNA elongation under the stretching force of 12 pN (Inset: schematic representation of the experimental scheme. A DNA tether is stretched with a controlled force by adjusting the distance between the beads using dual-trap optical tweezers). (**B**) DrRecA–ssDNA filaments reversibly switch between stretched and compressed conformations depending on the presence of ATP, which is characterized by dynamic change of the filaments length at stretching force of 2 pN in the ATP-containing buffer (gray sections) and ATP-free buffer (white sections). DrRecA: a central enzyme of recombinational DNA repair; ssDNA: single-stranded trails.

**Figure 2 ijms-21-07389-f002:**
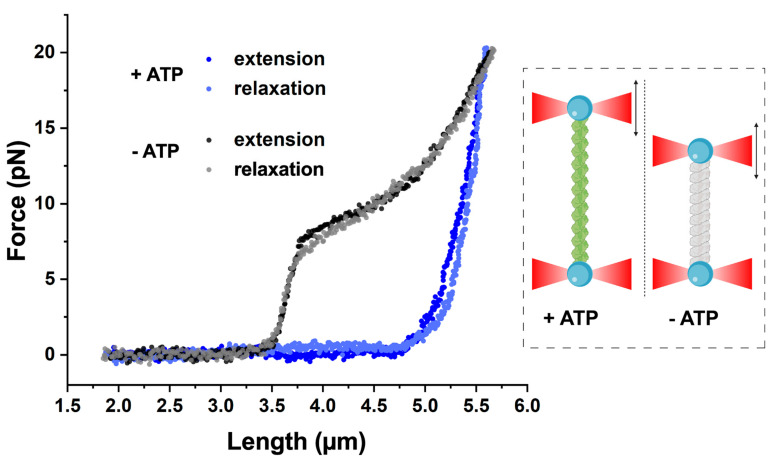
Force–extension behavior of DrRecA–ssDNA filaments in stretched (+ATP) and compressed (-ATP) states registered by increasing the distance between the beads while simultaneously recording the length of the tether and applied tension force.

**Figure 3 ijms-21-07389-f003:**
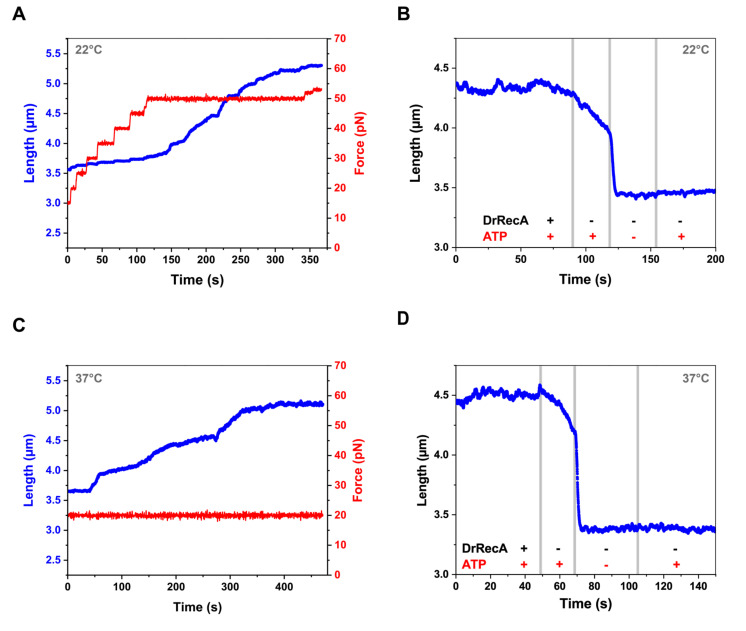
Dynamics of DrRecA–dsDNA filaments length. (**A**) dsDNA elongation upon DrRecA binding at 22 °C (polymerization was observed when applied force was 50 pN or higher). (**B**) Shortening of the DNA tether due to DrRecA dissociation from dsDNA while transferred first into the ATP-containing buffer without free DrRecA, and then in the ATP-free buffer under stretching force of 3 pN at 22 °C. Subsequent reintroduction back to the ATP-containing solution does not alter the DNA length. Gray vertical lines indicate moments the DNA tether was transferred between corresponding channels. (**C**) dsDNA elongation upon DrRecA binding at 37 °C and 20 pN stretching force. (**D**) same as (**B**) but at 37 °C. dsDNA: double-stranded DNA.

**Figure 4 ijms-21-07389-f004:**
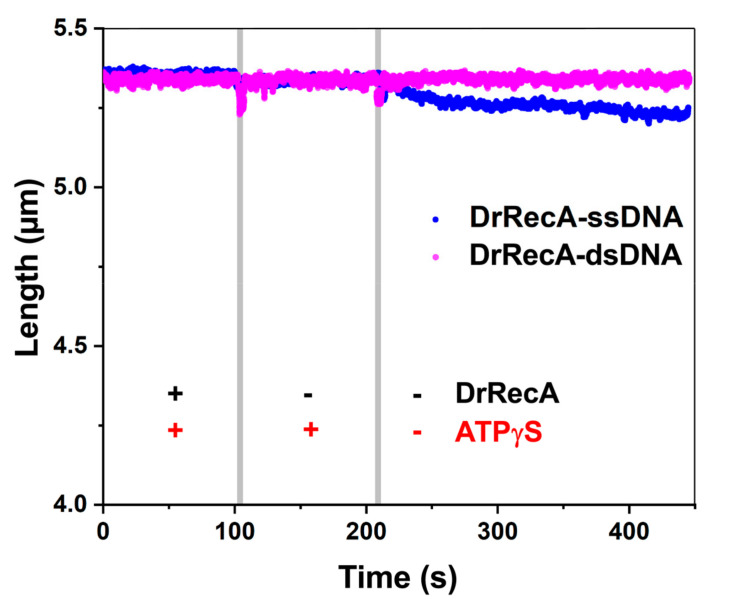
Dynamics of DrRecA filaments formed in the presence of ATPγS on ssDNA (blue) and dsDNA (magenta) under a stretching force of 3 pN, sequentially incubated in the channel containing both DrRecA and ATPγS, the channel with only ATPγS, and the channel lacking both DrRecA and ATPγS. Gray vertical lines indicate moments the DNA tether was transferred between corresponding channels.

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
