# Peer review of "Single-Molecule Insights into ATP-Dependent Conformational Dynamics of Nucleoprotein Filaments of Deinococcus radiodurans RecA"

_ijms, 2020, doi:10.3390/ijms21197389_

Round 1

Reviewer 1 Report

This is a re-submission of the earlier 903406 manuscript. Authors provides additional description in responding to the earlier-raised comments. Since there is no point-to-point response, this reviewer can only gauge from the highlighted text to see what have been modified in this version. Authors responded to some of my earlier comments (3 and 4). Unfortunately, the modified description didn’t respond to the comments in full, making it difficult to judge the validation of the conclusion.

(1, as in original comment 1): in the page 5 of the new version, the authors argued that no observed additional length change during 160-200 s (Figure 3B) confirmed that RecA dissociation in the absence of ATP. However, as the filament length was still decreasing during the 85-120 sec (didn’t reach a steady length yet), the fast length decrease could be resulted from the channel switching. The conclusive experiment should be done in (+RecA/+ATP) to (+RecA/-ATP), if the authors intend to establish that RecA dissociate from duplex DNA in the absence of ATP.

(2, as in original comment 2): F-x curve done in Figure 2 (RecA/ssDNA/ATP) is different from ref. 38, which used RecA/ADP state. Even with similar abrupt transition near 8-10 pN, F-x curve of E. coli RecA/ADP shown in ref. 38 was followed by a constant force plateau. This is different from the continuous force increase seen in Figure 2. In addition, extension and relaxation curves almost overlay, likely reflective of the similar protein-DNA complex during the pulling and relaxation. Therefore, it is difficult to justify that it represents the bare ssDNA.

Author Response

Comments and Suggestion for Authors:
This is a re-submission of the earlier 903406 manuscript. Authors provides additionaldescription in responding to the earlier-raised comments. Since there is no point-to-pointresponse, this reviewer can only gauge from the highlighted text to see what have beenmodified in this version. Authors responded to some of my earlier comments (3 and 4).
Unfortunately, the modified description didn’t respond to the comments in full, making itdifficult to judge the validation of the conclusion.

(1, as in original comment 1): in the page 5 of the new version, the authors argued that no observed additional length change during 160-200 s (Figure 3B) confirmed that RecA dissociation in the absence of ATP. However, as the filament length was still decreasing during the 85-120 sec (didn’t reach a steady length yet), the fast length decrease could be resulted from the channel switching. The conclusive experiment should be done in (+RecA/+ATP) to (+RecA/-ATP), if the authors intend to establish that RecA dissociate from duplex DNA in the absence of ATP.

Authors’ response: To justify that dissociation occurred in the presence of RecA and in the absence of ATP the experiment suggested by the reviewer was carried out. Results are presented in Figure S3, included in the revised Supplementary material. Elimination of ATP led to fast DrRecA dissociation from dsDNA even in the presence of free DrRecA in solution.

The following text was added to the manuscript:
Line 189: ‘Noteworthy, elimination of ATP led to fast DrRecA dissociation from dsDNA even in the presence of free DrRecA in solution (Figure S3)’.

Supplementary material:
Figure S3. Dynamics of the DrRecA-dsDNA filament length upon transition from the channel containing both DrRecA and ATP to the channel containing DrRecA and no ATP. During transition a constant tension of 3 pN was applied to the filament. Experiment was carried out at 22°C.

(2, as in original comment 2): F-x curve done in Figure 2 (RecA/ssDNA/ATP) is different from ref. 38, which used RecA/ADP state. Even with similar abrupt transition near 8-10 pN, F-x curve of E. coli RecA/ADP shown in ref. 38 was followed by a constant force plateau. This is different from the continuous force increase seen in Figure 2.

Authors’ response: Thank you for pointing this out. Indeed, despite the transition around 8-10 pN, the shape of the force-extension curve in ref.38 (ref.46 in the new version of the manuscript) differs from our measurements. It is worth to note that the experimental conditions used in that publication essentially differed from ours. Inactive EcRecA-ssDNA filaments in ref.38 were formed de novo in the presence of ADP. A force-extension curve measured by this way was followed by almost constant force plateau after abrupt transition near 8-10 pN. In our work an inactive DrRecA-ssDNA filament was obtained from the preassembled active DrRecA-ssDNA complex, the same way as it was done for EcRecAssDNA filament in ref.29 (new version of the manuscript), where the curve at higher forces representing a continuous force increase approaches the one of bare ssDNA, similar to our results. The difference in the shape of the curves might be due to the different experimental conditions used. The text has been edited (lines 159-169) to stress that similarity concerns the presence of an abrupt transition and the forces at which it occurs for both inactive EcRecA and DrRecA filaments on ssDNA.

In addition, extension and relaxation curves almost overlay, likely reflective of the similar protein-DNA complex during the pulling and relaxation. Therefore, it is difficult to justify that it represents the bare ssDNA.

Authors’ response: Figure S2 demonstrates that the same behaviour of DrRecA-ssDNA/(-ATP) filament remains upon stretching up to 50 pN and subsequent relaxation, indicating that the same protein-ssDNA complex remains before and after the stretching. Discussion on the possible nature of the transition was removed, the stress was added on the fact that similar DNA-protein complex remains throughout the pulling and relaxation cycle as suggested by the reviewer.

Following corrections were made:

Lines 159-169:

Original version:
‘However, at forces higher than 7.9 ± 0.7 pN (N = 9), a sharp change in the slope of the force-extension curve is observed, characterized by significantly enhanced extensibility of the tether, which resembles previously reported force-extension behaviour of the inactive EcRecA-ssDNA filament [29,38]. Further analysis of the force-extension behaviour at higher stretching forces revealed a close similarity to bare ssDNA (Figure S2) showing a forceinduced disruption of the filament structure. However, during the relaxation part the stiff filament structure was restored although at a little bit lower force but approximately the same end-to-end distance, indicating that the major part of DrRecA remained bound to ssDNA during the stretching-relaxation cycle.’

Edited version:
‘However, at forces higher than 7.9 ± 0.7 pN (N = 9), a sharp change in the slope of the force-extension curve is observed, characterized by significantly enhanced extensibility of the tether. An abrupt transition at similar stretching forces has been previously reported for the inactive form of EcRecA-ssDNA filaments [34,46]. However, unlike the constant force plateau around 8 pN measured for EcRecA-ssDNA filaments in the presence of ADP [46], stretching of inactive DrRecA-ssDNA filaments at forces higher than 8 pN resulted in a continuous force increase, approaching a force-extension behaviour of bare ssDNA (Figure  S2) similar to [34]. Although the nature of such force-induced transition cannot be resolved
solely based on force-extension analysis, extension and relaxation of the inactive DrRecAssDNA filament follow almost identical curves, likely reflecting that similar DNA-protein complex exists during the stretching-relaxation cycle.

Reviewer 2 Report

The authors have suitably addressed the issues raised during my first review.

Author Response

Issues raised by the Reviewer 2 have been already addressed.

Reviewer 3 Report

Overall, it is a very well written manuscript with adequate experimental data with meticulous discussion to conclude the behavior of DrRecA-ssDNA filaments.

However, adding a data on the behavior of DrRecA-ssDNA and dsDNA filaments in presence of ADP and also switching alternatively between ATP and ADP would have given more insight into the dynamic behavior of DrRecA filaments. It is just a thought!

If the DrRecA nucleofilament behavior on ssDNA is very similar to the EcRecA nucleofilaments how does DrRecA prefers reverse strand-exchange as oppose to EcRecA, and why don’t they functionally complement? Definitely, I am not expecting authors to address these questions in the manuscript, however, these questions have become more puzzling and also very intriguing!

Minor comments:

I personally wouldn’t prefer call RecA, Ras51, Dmc1 as recombinases! In my opinion, Recombinases are those who perform recombination all by themselves (recognize, cut and replace a region based on a specific sequence by single handedly; for example: Cre recombinase!) I would rather prefer to call RecA, Rad51, Dmc1 proteins as DNA-strand exchange proteins, it is what they do based on homology.

In line # 35, replace ‘accurate’ with ‘error-free’.

In line # 42, replace ‘Synthesis of S. flexneri RecA’ with ‘Expression of Shigella flexneri RecA’

In Figure 1, adding a schematic representation of microfluidic cell with various channels describing reaction condition would be useful to the readers.

Reviewer 4 Report

The authors show an interesting work about DrRecA. It is known that DrRecA bind dsDNA with more affinity that ssDNA contrary to all the studied RecA/RadA/Rad51 proteins. Here, the most striking point is to see that depending the hydrolysis of ATP, DrRecA is able to bind efficiently ssDNA.

I have only minor points to adress to authors:

Introduction is interesting, however, only D. radiodurans is mentionned whereas RecA from other Deinococcales were studied, in particular in D. deserti since mutants are also affected and RecA is involved in this species into the UV mutagenesis that were never found into D. radiodurans.

Similarly, RadA, the homologous recombination protein in Archaea is not detailed into the introduction, however, which of Sulfolobus or Thermococcus are well known and studied.

Line 42: authors wrote "in recA-defective", recA should be in italic if they speak about the gene as I imagine.

Line 98: authors explained that experiment were done at 22°C but later in the article experiment on dsDNA was also done at 30°C. Since Deinococcus grows principally at 30°C, why the experiment was not also tested at 30°C ?

Round 2

Reviewer 1 Report

All comments are addressed.

This manuscript is a resubmission of an earlier submission. The following is a list of the peer review reports and author responses from that submission.

Round 1

Reviewer 1 Report

This is on the surface a well-executed single molecule study of the effects of nucleotides on the filaments made by the Deinococcus RecA protein. This is an interesting protein and the comparison with the E coli protein is warranted. The results are mainly straightforward. The problem comes in the interpretation. The conclusion put forward is that ATP hydrolysis by the DrRecA protein results in an accordion-like back and forth transition between active (extended) and inactive (compressed) conformations. This is a model proposed some time ago for the coli protein, and it has been thoroughly disproven. A wider search of the literature will demonstrate this. As with the early data used to support this idea for the coli protein, the new data does not at all demonstrate such a mechanism. The problem can be seen in Figure 1B. During the 30 seconds or so that ATP is present, there are multiple ATPs being hydrolyzed by every RecA in every filament. The length of the filament is the length DURING ATP hydrolysis. As individual RecA monomers bind and hydrolyze ATP, there is no effect on filament length because the individual monomers are not hydrolyzing ATP synchronously. When ATP is removed, ATP hydrolysis of course halts and the filament collapses. However, under normal circumstances ATP hydrolysis by individual RecA monomers in the filament is not at all synchronous. Filament length is a constant reflecting a steady state in which a few monomers are hydrolyzing ATP at any given time. There is potential in this work, but propagating an incorrect model of how ATP hydrolysis is coordinated in a RecA filament would be a bad idea.

Reviewer 2 Report

Alekseev et al. use single-molecule optical tweezers to study the properties of nucleoprotein filaments formed by Deinococcus radiodurans RecA recombinases in the presence of ss and dsDNA. They observed the different stretching properties in the presence of ssDNA and dsDNA and furthered suggested the dynamic difference in the presence of different nucleotide states. The authors argued that these differences contribute to the biochemical properties, which might moonlight the preference of DrRecA to use DrRecA-dsDNA filament to execute the strand exchange reactions.

Earlier biochemical work has shown that EcRecA can form different nucleoprotein filaments in the presence of ATP and ADP (or no nucleotide). Optical tweezers experiments could have provided more mechanistic details about the origins of these differences. Unfortunately, results shown in this manuscript didn’t provide sufficient information to further our molecular understanding on this interesting and special enzyme. Figures and figure captions are not written in consistence, which made it challenging to appreciate this work.

Major concerns:

(1) The filament length change could be resulted from (i) different nucleotide states (different binding conformations) or (ii) dissociation of DrRecA from the filament (i.e. koff). When proteins are removed from the reaction mixture, it becomes difficult to conclusively confirm whether the reduced length is indeed resulted from the different conformations. For example, data from Figure 3B/D, can also be explained as additional DrRecA dissociation.

(2) Why F-x curve of DrRecA-ssDNA/(-ATP) in figure 2 shows an abrupt transition around 8 pN? If the extension and relaxation curves overlay, it seems to suggest that the same protein-ssDNA complex remains before and after the stretching. More control experiments are required to explain this F-x curve. It is also useful to show the force data up to 40-50 pN. In addition, please show F-x curve for the pure ssDNA substrates.

(3) Figure caption description in Figure 3 does not match the data shown in the figure.

(4) There is no methods and materials section for the preparation of the ssDNA, nor the validation of the ssDNA substrates.

Reviewer 3 Report

The manuscript entitled "Single-molecule insights into ATP-dependent conformational dynamics of nucleoprotein filaments of Deinococcal radiodurans RecA" presents a single-molecule study using optical tweezers of RecA polymerization and structural dynamics on ss- and dsDNA in the presence of nucleotides (ATP and its non-hydrolyzable analogue, ATPγS).

The manuscript is well written, presents well performed experiments and provides insight into the conformational flexibility of Deinocoocus radiodurans RecA on DNA. The conclusions are supported by the results; no overstatements or overinterpretation of the data.

This manuscript is suitable for publication after minor revision.

The main point which should be addressed is the lack of discussion of earlier DNA binding data described in the introduction, when discussing the different behaviours of DrRecA on ss- and dsDNA in these single molecule measurements. Notably, the authors should comment on the fact that in this work, DrRecA does not efficiently assemble on dsDNA, while earlier work indicated that binding of DrRecA to short dsDNA fragments was one magnitude tighter than to ssDNA (ref 21).